# A new set of indicators for model evaluation complementing to FAIRMODE's MQO

Alexander de Meij[1], Cornelis Cuvelier[2]♦, Philippe Thunis[2], Enrico Pisoni[2]

1MetClim, Varese, 21025, Italy
2 European Commission, Joint Research Centre (JRC), 21027, Ispra, Italy
♦retired with Active Senior Agreement

*Correspondence to*: Philippe Thunis (philippe.thunis@ec.europa.eu)

**Abstract.** In this study, we assess the relevance and utility of several performance indicators (Model Quality [bias] and Model Performance [temporal and spatial] Indicators, developed within the FAIRMODE framework by evaluating eight CAMS models and their ensemble in calculating concentrations of key air pollutants, specifically $NO_2$, $PM_{2.5}$, $PM_{10}$, and $O_3$. The models' outputs were compared with observations that were not assimilated into the models. For $NO_2$, the results highlight difficulties in accurately modelling concentrations at traffic stations, with improved performance when these stations are excluded. While all models meet the established criteria for $PM_{2.5}$, indicators such as bias and Winter-Summer gradients reveal underlying issues in air quality modelling, questioning the stringency of the current criteria for $PM_{2.5}$. For $PM_{10}$, the combination of MQI, bias, and spatial-temporal gradient indicators prove most effective in identifying model weaknesses, suggesting possible areas of improvement. $O_3$ evaluation shows that temporal correlation and seasonal gradients are useful in assessing model performance. Overall, the indicators provide valuable insights into model limitations, yet there is a need to reconsider the strictness of some indicators for certain pollutants.

## 1. Introduction

Air Chemistry Transport Models (ACTMs) are used to calculate the complex physical and chemical processes that play a role in the formation and removal of gases and aerosols (e.g. $NO_2$, $O_3$, $SO_x$, PM) from our atmosphere. Also, an ACTM is an instrument to assess the effects of future changes in aerosol (+ precursor) emissions, and models are therefore used to assist policy making in the design of effective reduction strategies to improve the air quality.

An ACTM requires a set of input data (e.g. emission and meteorology) and a description of (dynamical and chemical) processes to calculate gas and aerosol pollutants. The description of these processes in the model is associated with uncertainties. Model performance depends on the quality of the input data (e.g. emission and meteorology) and on the way we represent the dynamical and chemical processes leading to gas and aerosol concentrations. Many approaches exist to manage these two points, leading to some variability among model results. This variability can be understood as the modelling uncertainty. Previous studies investigated the uncertainties associated with certain processes when air chemistry transport models are used, through model ensemble approach such as described in Vautard et al., (2006, 2009), Van Loon et al., (2007). Other studies investigated the uncertainties associated with model resolution (De Meij et al., 2007, Wang et al., 2015, Huang et al., 2022), chemistry (Textor et al., 2006, Thunis et al. 2021b, Clappier et al., 2021), meteorology (De Meij et al., 2009, Gilliam et al., 2015) and emission inventories (Thunis et al., 2021c, Colette et al,

2017). Over the years, air quality modelling has improved as model's uncertainties have been reduced. Often classical statistical parameters are used to evaluate the ACTM's capability in calculating air pollutants. For example, bias (measure of overestimation or underestimation), standard deviation (a measure of the dispersion of the observed/calculated values around the mean), temporal correlation coefficient (linear relationship between model and observations), root mean square error (a measure of difference between the model and the observations; measure of accuracy) to name a few. In the United States of America (USA), modeling guidance and performing evaluation was firstly introduced by the US Environmental Protection Agency (EPA) in 1991. Followed by introducing the concepts of "goals" (i.e. model accuracy) and "criteria" (i.e. threshold of model performance) in studies by Boylan and Russell (2006) and Emery et al. (2017). In the USA, air quality models are evaluated based on several model performance indicators to ensure their accuracy and reliability. These indicators are: Mean Bias (MB), Mean Absolute Error (MAE), Root Mean Square Error (RMSE), Fractional Bias (FB), Normalized Mean Bias (NMB), Normalized Mean Error (NME), Pearson Correlation Coefficient (R or R2) and Index of Agreement (IOA). For operational air quality performance, additional indicators are used: Prediction Accuracy, Hit Rate & False Alarm Rate and Skill Scores. The EPA has specific Regulatory Performance Criteria for key pollutants like PM2.5, NO2 and O3.

For O3 modeling a model is considered acceptable if:

•        NMB is within ±15%

•        NME is ≤ 25%

For PM2.5 the performance goals are:

•        NMB within ±30%

•        NME ≤ 50%

Also, EPA's Support Center for Regulatory Atmospheric Modeling (SCRAM) provides resources and guidance on air quality models and their evaluation.

In China, Huang et al. (2022) proposes benchmarks for MB, MAE, RMSE, IOA, R and FB for air quality model applications since there are no unified guidelines or benchmarks developed for ACTM applications in China. Huang et al. (2022) methodology is based on Emery et al., (2017), applying goals and criteria for NMB, NME, FB, FE, IOA and R. Note that the model criteria are fixed in Huang et al., (2021) and Emery et al., (2017), while in our work the criteria depends on the observation uncertainties, which is for each pollutant different.

These indicators are in general used to assess the model's performance against measurements. However, these indicators do not tell whether model results have reached a sufficient level of quality for a given application. In Huang et al., (2022) recommendations are given to provide a better overview of model performance. For example, for PM2.5 the NMB should be within 10 % and 20 % and R should lay between 0.6 and 0.7 for hourly and daily PM2.5 and between 0.70 and 0.90 for monthly PM2.5 concentration values, Also, different temporal resolutions for PM2.5 calculated values are introduced. Furthermore, benchmarks for speciated PM components (elemental/organic carbon, nitrate, sulphate and ammonium) were recommended.

Along the same line, the Forum for Air quality Modelling (FAIRMODE) (https://fairmode.jrc.ec.europa.eu/home/index) developed several specific quality assurance and quality control (QA/QC) indicators and associated a threshold to each of them, that indicates the minimum level of quality to be reached by a model for policy use (Janssen and Thunis, 2022). Recent studies that have used these QA/QC indicators and associated thresholds to evaluate ACTM's performances are Kushta et al., (2018) and Thunis et al., (2021a).

Note that the goals and criteria proposed in the US or in China remain independent of the concentration level. In this work, we define a threshold on the maximum accepted modelling uncertainty. Because we do not know the modelling uncertainty in practice, we set it proportional to the measurement uncertainty. With this definition, the more uncertain the measurement is (e.g. relative uncertainties become larger in the lower concentration range), the more flexibility we allow to the modelling results, i.e. a higher threshold value (and vice-versa).

The goal of this study is to assess the relevance and usefulness of FAIRMODE's model quality assessment indicators and FAIRMODE's QA/QC Tools, by using as benchmark the Copernicus Atmospheric Monitoring Service (CAMS) air quality modelling and ensemble results over Europe.

More details on the models, methodology and emission inventories are given in Chapter 2. Followed by the analysis of the results in Chapter 3. In Chapter 4 the conclusions are provided.

## 2.	Methodology

CAMS produces annual air quality (interim) re-analysis for the European domain at a spatial resolution of 0.1 x 0.1 degrees. A median ensemble is calculated from individual outputs, since ensemble products yield on average better performance than the individual model products. The spread between the eight models can be used to provide an estimate of the analysis uncertainty (Marécal et al., 2015, CAMS, 2020).

We assess the relevance and usefulness of FAIRMODE's model quality assessment indicators by means of evaluating simulated air pollutants ($NO_2$, $O_3$, $PM_{2.5}$ and $PM_{10}$) by the eight CAMS models for the year 2021, by comparing with observational data from the European Air quality database and assess the results against specific performance indicators. The evaluation of the model's performance is based on the comparison with observations that are not used to assimilate calculated concentrations. The eight CAMS models are:

CHIMERE (FR), DEHM (DK), EMEP (NO), FMIA-SILAM (FI), GEMAQ (PL), KNMA-LOTUS-EUROS (NL), MFM-MOCAGE (FR), RIU-EURAD-IM (DE) and Ensemble (ENSKCa). The CAMS regional air quality models generate reanalysis, detailing the concentrations of major atmospheric pollutants in the lowest layers of the atmosphere across the European domain (ranging from 25.0°W to 45.0°E and 30.0°N to 72.0°N). The horizontal resolution is approximately 0.1°, varying from around 3 km at 72.0°N to 10 km at 30.0°N.

 For that reason, an overview of the type of assimilation methodology, which species are assimilated, together with gas and aerosol schemes are given in Table S1 of the Supplement material. More details of the different models are described in

(https://confluence.ecmwf.int/display/CKB/CAMS+Regional%3A+European+air+quality+reanalyses+data+documentat ion). The data can be downloaded here: https://atmosphere.copernicus.eu/data

For the statistical analysis, the FAIRMODEs' benchmarking methodology is applied, that provides many different statistical parameters, which are described in FAIRMODE's Guidance document (Janssen et al., 2022).

The indicators and modelling criteria described in this study, were defined in the context of FAIRMODE to support the application of modelling in the context of the Air Quality Directive.

Initially, FAIRMODE developed a single model performance indicator: the MQI. While this indicator provides relevant pass/fail test, passing the test does not ensure that modelling results are fit for purpose. This is why additional indicators have progressively been added, in particular to assess how models capture temporal and spatial aspects. The Modelling Quality Indicator (MQI) is a statistical indicator of the accuracy of a specific modelling application calculated based on measurements and modelling results. It is defined as the ratio between the model-measured bias at a fixed time (i) and a quantity proportional to the measurement uncertainty as:

$$MQI(i) = \frac{|O_i - M_i|}{\beta U(O_i)} \tag{1}$$

Where $U(O_i)$ is the measurement uncertainty and β a coefficient of proportionality. The normalisation of the bias by the measurement uncertainty is motivated by the fact that both model and measurements are uncertain. We want to account for the fact that when measurement uncertainty is large, some flexibility on the model performance can be accepted, translating in accepting larger model-observed errors. With a current value of 2 proposed for β, the quality of a modelling application is said to be sufficient when the model-observation bias is less than twice the measurement uncertainty.

Applied to a complete time series, Equation (1) can be generalized to:

$$MQI = \frac{RMSE}{\beta RMS_U} \tag{2}$$

A complete time series entails 75% data availability over the selected time period. Note that this number is less than the one requested in the Ambient Air Quality Directive (AAQD, 2024) (i.e. 90%) to increase the available number of measurement stations for validation. We however impose that available data are representative of the full year.

With this formulation, the RMSE between observed and modelled values (numerator) is compared to the root mean square sum of the measurement uncertainties (RMSU) which value is representative of the maximum allowed measurement

uncertainty (denominator).

For yearly averaged pollutant concentrations, the MQI formula is adapted so that the mean bias between modelled and measured concentrations is normalised by the uncertainty of the mean measured concentration ($U(\bar{O})$):

$$MQI = \frac{|\bar{O} - \bar{M}|}{\beta U(\bar{O})} \tag{3}$$

More details on formulation (1), (2) and (3) can be found in the Modelling Quality Objective (MQO) guidance document (Janssen et al., (2022).

For the statistical analysis of the four air pollutants, we use for $NO_2$ the hourly values and for $O_3$ the 8-hour running mean maximum values. Whilst for $PM_{2.5}$ and $PM_{10}$ the daily averages are used. These different time intervals are in compliance

with the EU air quality standards as stated in the AAQD. The time intervals are specific for each air pollutant, because the observed health impacts associated with the various pollutants occur over different exposure times.

The MQO is fulfilled when the MQI is less or equal to 1., for at least 90% of the available stations. The yearly MQI is in general more challenging to fulfil than the daily MQI (but this is not a rule), because of the smallest measurement uncertainties for yearly mean observed concentrations. The underlying reason for this is that the impact of random noise

and periodic re-calibration on the daily observations lead to larger uncertainties, which are compensated for yearly averages.

The main drawback of the MQOs is that they provide a single summary pass/fail information for a modelling application. This simple test does not prevent a modelling application to pass for the wrong reason under certain circumstances. In addition, it does not provide any information on the capability of the model to reproduce hot spot areas (spatial variability)

or on the timing of the pollution peaks (temporal variability).

For these reasons, additional indicators are proposed to assess the capacity of models to capture the temporal and spatial variability of the measurements. These indicators are based on temporal and spatial correlation or standard deviations that are normalised by the measurement uncertainty.

These indicators are constructed as follows:

For hourly frequency model output, values are first yearly averaged at each station. A temporal or spatial correlation and standard deviation indicator are then calculated for this set of values. The two indicators are normalised by the measurement uncertainty of the average concentrations:

$$RMS_{\bar{U}} = \sqrt{\frac{1}{N}\sum U(\bar{O})^2} \tag{4}$$


The same approach applies for yearly frequency output.

These indicators are defined in Table 1.

Table 1 Model performance indicators for temporal and spatial correlation.

| | Model Performance Indicator (MPI) | Model Performance Criteria (MPC) |
|---|---|---|
| Correlation (5) | $$MPI = \frac{1 - R}{0.5\beta^2 \frac{RMS_{\overline{U}}^2}{\sigma_O \sigma_M}}$$ $(BIAS = 0, \sigma_O = \sigma_m)$ | $MPC: MPI \leq 1$ |
| Standard deviation (6) | $$MPI = \frac{|\sigma_M - \sigma_O|}{\beta RMS_{\overline{U}}}$$ $(BIAS = 0, R=1)$ | |


Where the Model performance criteria is the criteria to be fulfilled in order to reach the quality objective of the modelling application.

On top of these already agreed indicators included in FAIRMODE MQI system approach, we propose to complement
them with incremental indicators, where relevant[1], to assess how concentration gradients between rural and urban or between traffic and urban stations are reproduced by the model. This is relevant in the context of the Ambient Air Quality Directive (AAQD), because the design of the monitoring network aims to capture existing gradients and differences occurring as a result of different pollution sources and different dispersion situations. These additional spatial indicators can be constructed similarly to other MQIs, i.e. normalised by the measurement uncertainty.
For example, the modelled incremental change between rural background (RB) and urban background (UB) locations is defined as:

$$INC_{UB-RB}^{model} = \overline{M}_{UB} - \overline{M}_{RB} \qquad (7)$$

where M is the model value and similarly for the measured increment:
$$INC_{UB-RB}^{observed} = \overline{O}_{UB} - \overline{O}_{RB} \qquad (8)$$

These indicators are then normalised by the measurement uncertainty, see Table 2.

Table 2. Model performance indicators that describe the incremental change between rural background (RB) and
background (UB) locations.

| | Model Performance Indicator (MPI) | Model Performance Criteria (MPC) |
|---|---|---|
| UB – RB (9) | $MPI = 1/\beta * \frac{INC_{UB-RB}^{model} - INC_{UB-RB}^{observed}}{0.5 * (RMS_{\overline{U(UB)}} + RMS_{\overline{U(RB)}})}$ | MPC: MPI $\leq$ 1 |

---

[1] Indicators can only be applied with models that are designed to simulate the station types that are used in the indicators (e.g. urban-traffic incremental indicators cannot be applied to models that only simulate background levels).

| UT – UB (10) | $$MPI = 1/\beta \, * \frac{INC^{model}_{UB-UT} - INC^{observed}_{UB-UT}}{0.5 * (RMS_{\overline{U(UB)}} + RMS_{\overline{U(UT)}})}$$ | |

where UT stands for "urban traffic".

As mentioned earlier, the MQO generally applies to the average of a specific period, currently, one year. Consequently, it provides no information whether the modelling application manages to capture the temporal variability of the air quality situation. Since the AAQDs include also in the assessment the evaluation of exceedances for specific temporal indicators, the capability of the modelling application to reproduce the temporal variations becomes highly relevant in the context of air quality management.

For that reason, additional indicators to assess the temporal coherence of model results, at different frequencies are provided (Table 3). These include seasonal, week/week-end or day/night indicators. Measurement and modelling results are then aggregated (all stations belonging to a certain type (urban – rural –traffic – industrial) together and checks are made through the following indicators:

Table 3 Model quality indicators at different frequencies: seasonal, week/week-end or day/night.

| | | Model Performance Indicator (MPI) | Model Perf. Criteria (MPC) |
|---|---|---|---|
| Seasonal (12) | Industry | $$MPI = \frac{SeasDiff^{mod}_{Ind} - SeasDiff^{obs}_{Ind}}{\beta RMS_{\overline{U}}}$$ | |
| | Traffic | $$MPI = \frac{SeasDiff^{mod}_{traffic} - SeasDiff^{obs}_{traffic}}{\beta RMS_{\overline{U}}}$$ | |
| | Background | $$MPI = \frac{SeasDiff^{mod}_{bg} - SeasDiff^{obs}_{bg}}{\beta RMS_{\overline{U}}}$$ | |
| Week / weekend (13) | Industry | $$MPI = \frac{WeekDiff^{mod}_{Ind} - WeekDiff^{obs}_{Ind}}{\beta RMS_{\overline{U}}}$$ | MPC: MPI ≤ 1 |
| | Traffic | $$MPI = \frac{WeekDiff^{mod}_{traffic} - WeekDiff^{obs}_{traffic}}{\beta RMS_{\overline{U}}}$$ | |
| | Background | $$MPI = \frac{WeekDiff^{mod}_{bg} - WeekDiff^{obs}_{bg}}{\beta RMS_{\overline{U}}}$$ | |
| Day/night (14) | Industry | $$MPI = \frac{DayDiff^{mod}_{Ind} - DayDiff^{obs}_{Ind}}{\beta RMS_{\overline{U}}}$$ | |
| | Traffic | $$MPI = \frac{DayDiff^{mod}_{traffic} - DayDiff^{obs}_{traffic}}{\beta RMS_{\overline{U}}}$$ | |

| | Background | $$\text{MPI} = \frac{DayDiff_{bg}^{mod} - DayDiff_{bg}^{obs}}{\beta RMS_{\overline{U}}}$$ | |
|---|---|---|---|

The Air Quality Directive of the European Commission provides definitions for different types of air quality monitoring stations based on their location and the pollution sources they are exposed to. These station types ensure a comprehensive assessment of air quality across different environments, helping policymakers and researchers analyze pollution trends and enforce regulatory limits. The key definitions are:

- **Traffic stations**: Near major roads or intersections (at least 25 meters from major intersections, but no more than 10 meters from the road), dominated by vehicle emissions ($NO_2$, $PM_{10}$, $PM_{2.5}$), reflecting population exposure to road transport pollution.

- **Urban stations**: In residential or commercial areas (more than 50 meters away from major roads and more than 4 km away from industrial sources), measuring background pollution levels affecting the general urban population.

- **Industrial stations**: Near factories or power plants, monitoring emissions like $SO_2$, $NO_2$, heavy metals, and VOCs.

- **Rural stations**: In the countryside or suburban areas (at least 20 km from urban areas and 5 km from industrial sources), assessing regional and long-range pollution transport.

A more detailed description of the station types can be found in Annex III ("Assessment of Air Quality and Location of Sampling Points") of the Air Quality Directive (2008/50/EC).

# 3.     Results

To best visualize all these indicators, we use a graphical representation in terms of radar plots. These plots help to assess the relevance and usefulness of the different statistical indicators by comparing all of them in a single diagram. We use this approach to assess models' performance for Spain, France, Germany, Poland and Italy. This allows us to see if (1) the MQI values fulfil the MQO. If this is not the case, the radar plots help to understand which of the other indicators are useful in determining the model's skill through analysing (2) the temporal and spatial indicators (1-R and Stdev), followed by (3) studying the models' capability in calculating the temporal variability i.e. seasonal (Winter-Summer [W-S]), week-weekend (Wk-We) and day-night (D-N) indicators and spatial indicators (e.g. urban background - rural background gradient).

## 3.1     Model performance analysis for $NO_2$

In Fig. 1, the statistics for $NO_2$ are shown for (a) Spain, (b) France, (c) Germany, (d) Poland and (e) Italy by all models considering all stations (i.e. background (B), urban, traffic (T), industry (I)). The green circle represents the reference line, that is MQI is 1.0. Results for any statistical parameter that fall within the circle indicates that the MQO is achieved. Anything that falls outside the green circle indicates a poor agreement of the model results when compared to observations. The cyan solid contour in each radar plot represents the Ensemble Median. The other ACTM's are presented by different colours.

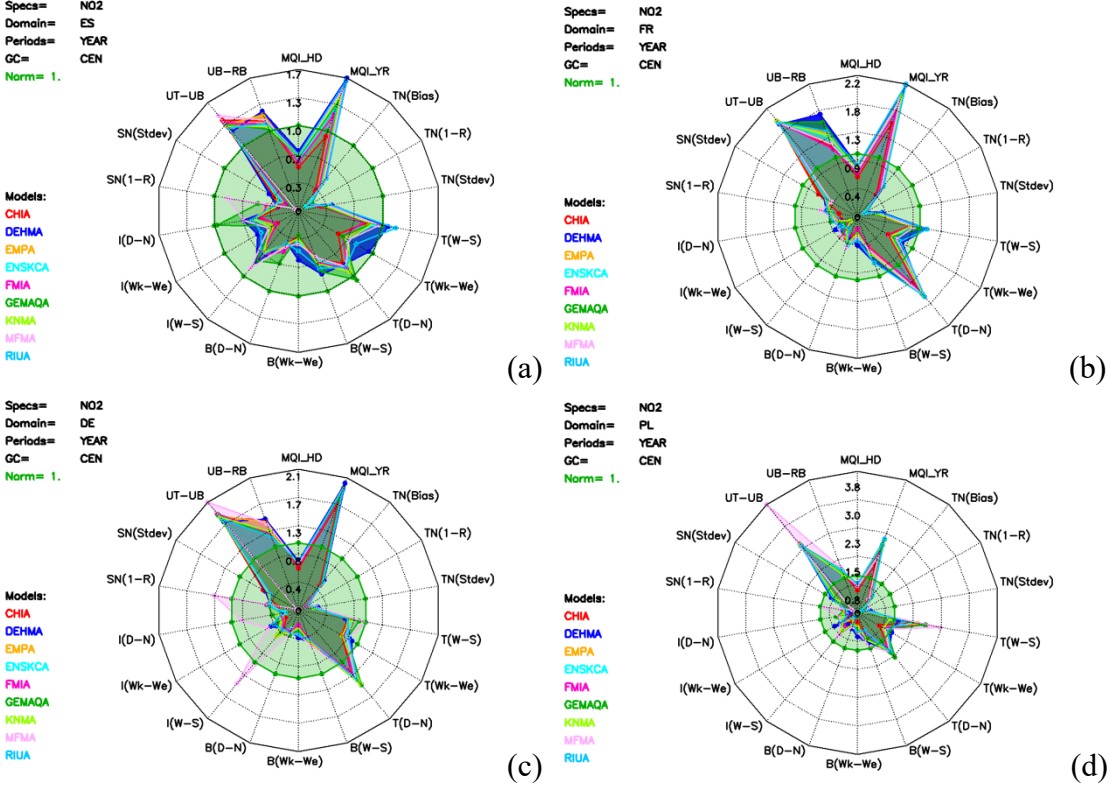

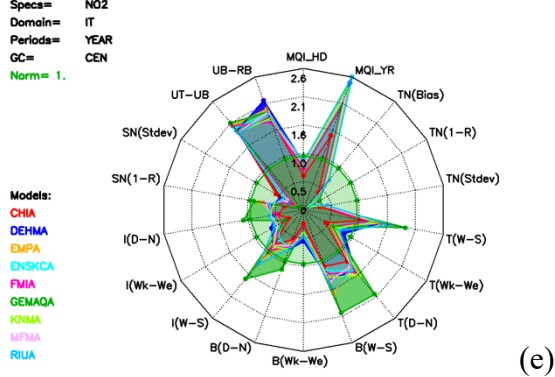

**Figure 1. Radar plots of the calculated air quality model indicators for NO₂ for different countries: (a) Spain, (b) France, (c) Germany, (d) Poland and (e) Italy. Indicators are: MQI Hourly (MQI_HD), MQI Year (MQI_YR), Bias, 1-R (Time), Standard deviation (Time), gradients for Winter-Summer, Week-Weekend, Day-Night for Traffic, Industry, Background (T, I, B), 1-R Spatial, Standard Deviation spatial, Yearly Urban-Traffic vs Urban-Background (Year UT-UB), Yearly Urban-Background vs Rural-Background (Year UB-RB).**

Fig. 1 shows that the yearly MQIs (MQI_YR) are generally higher than 1.5 for all models and all countries (a) Spain, (b) France, (c) Germany, (d) Poland and (e) Italy, indicating that the MQOs are not achieved, while the short-term MQIs (MQI_HD) fulfil the MQOs. As mentioned earlier, the yearly MQI is more difficult to fulfil than the daily MQI, because of smaller measurement uncertainties for yearly mean observed concentrations. As a consequence, the MQI_YRs values are higher than MQI_HD, indicating that each model has difficulties capturing well the observed yearly concentrations for NO2.

As mentioned earlier, the MQOs tells if the model fails or passes the MQI, but with limited information on the model's capability to calculate the temporal and spatial variability of the air pollutant concentrations. This is why we introduced additional indicators, see (Equations 4 – 6), which present the bias and temporal- and spatial correlation.

A more stringent source of information to the additional indicators in Equations 4 – 6 are presented in Equations 7 - 10. We see that for example these indicators describe the differences between biases for Day versus Night values for Background [B(D-N] and Industry [I(D-N] stations are smaller than 1.0, except for Italy by GEMAQA (see Annex). Therefore, one would expect that the models are, in general, capable of calculating well the NO₂ concentrations. But when the spatial indicators are considered, this is clearly not the case. For example, the spatial concentration gradient around a Traffic station considering the Urban Background stations (Year UT-UB) and UB-RB (concentration gradient around a Background station considering Rural Background stations), exceeds the reference line (1.0) indicating that the model's capability in calculating the spatial gradient is poor when compared to the observations and therefore doesn't fulfil the MQO. This highlights the value of these indicators in assessing model performance.

This can be explained by the fact that the model resolution (0.1 x 0.1) is too coarse to capture the emissions from the road transport sector. This is illustrated in Fig. 2, which shows the difference between observations and calculated yearly mean NO₂ concentrations for Traffic, Industry, All and Background stations for Germany. The calculated NO₂ concentrations for Traffic and All stations remain flat, i.e. the concentrations are very similar around 13 μg/m³. While the difference in observed concentrations (grey bar) between Traffic stations and All stations is around 7 μg/m³ (27 for Traffic and 20 μg/m³ for All stations).

Also, the Bias for Traffic stations is much larger (up to -14 μg/m³), while the Bias for all stations is smaller (up to -9 μg/m³), see Fig. 3. This indicates that the models have difficulties in calculating the NO₂ concentrations for Traffic

stations as mentioned earlier. Once again this is expected, given the resolution of the models, but it shows the relevance of the indicators and associated thresholds to detect it.

The mean calculated $NO_2$ concentrations by the models for Industry and Background stations agrees well with the observations. This reflects into low bias for Industry and Background stations ($< 3$ μg/m$^3$).

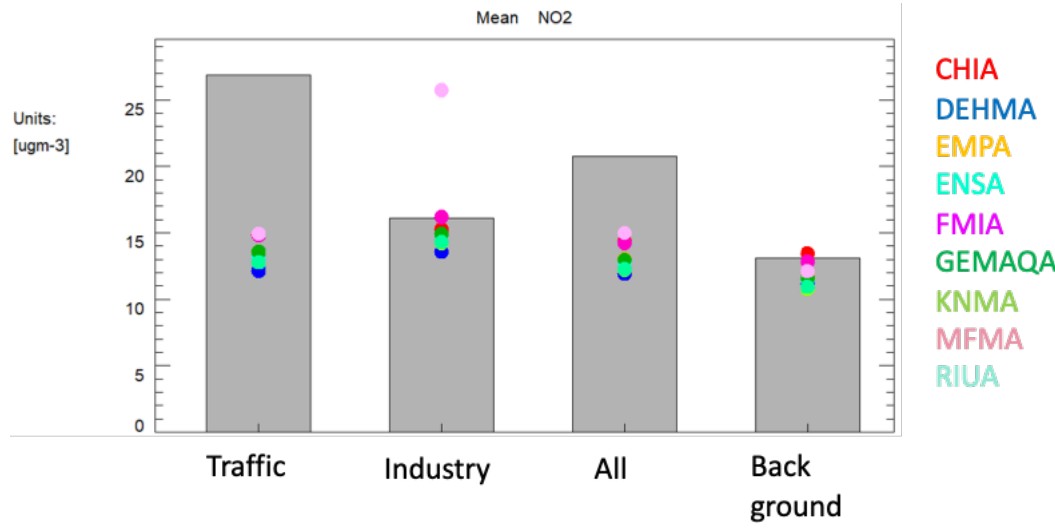

**Figure 2. Yearly mean observed (grey bar) and calculated (coloured dots) NO₂ concentrations for Germany for Traffic, Industry, All and Background stations.**

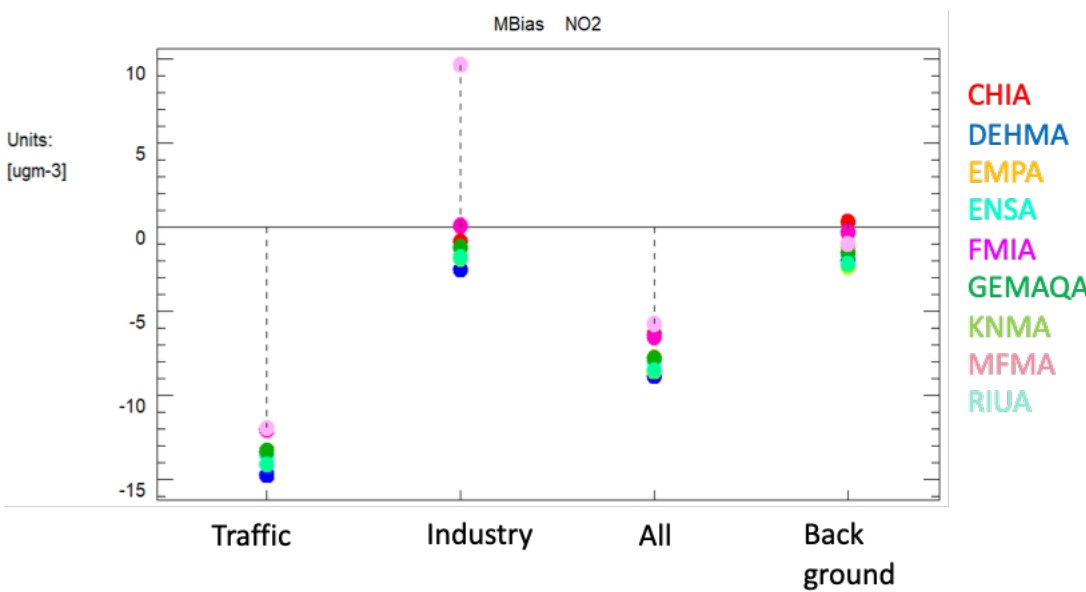

**Figure 3. Yearly mean bias for NO₂ for Traffic, Industry, All and Background stations for the different models (coloured dots) for Germany stations.**

Looking in more details we show in Fig. 4 the comparison between the model versus Day - Night and Winter - Summer mean observations for Traffic and Background stations in Italy. Well behaving results should lie along the 1 to 1 line. Results located in the lower right and upper left parts of the graphs are poor.

Like the other models, GEMAQA (Fig. 4a) shows a poor agreement for the traffic stations to capture the Day - Night and Winter - Summer profiles for Italy. A similar behaviour is found for the Background stations as shown in Fig. 4b for

RIUA. Note that for the other countries the Day - Night and Winter - Summer profiles are satisfactory for Background stations, but not for Traffic stations. In general, for background stations, all indicator values remain below the threshold of 1.0, except for the GEMAQ model in Italy. This suggests that the models perform better in less complex environments and that these indicators may be less effective for assessing model performance in this context.

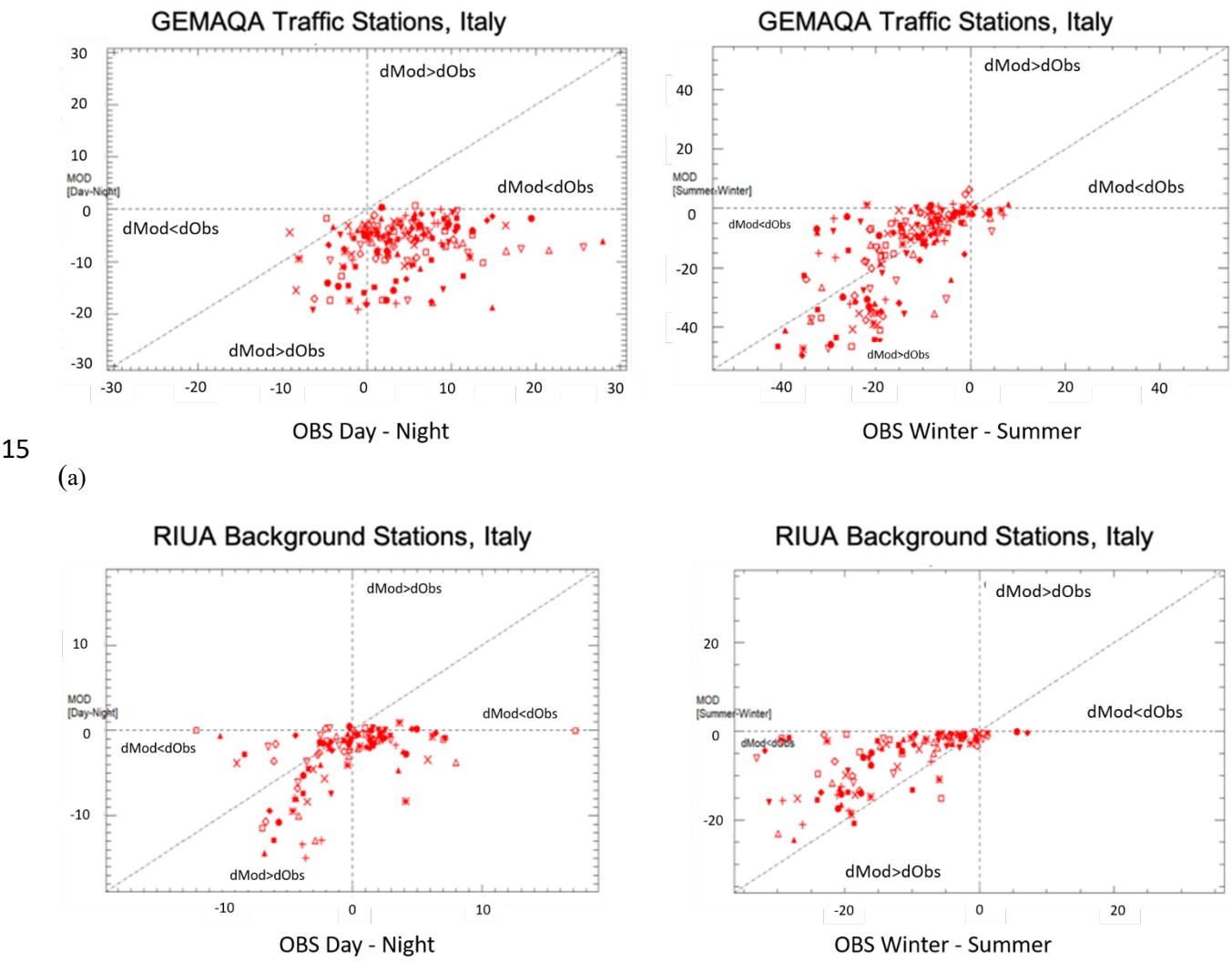

(a)

**Figure 4. NO$_2$ scatter plots of modelled versus observed day-night and summer-winter mean differences for traffic stations by (a) GEMAQA and background stations by (b) RIUA model.**

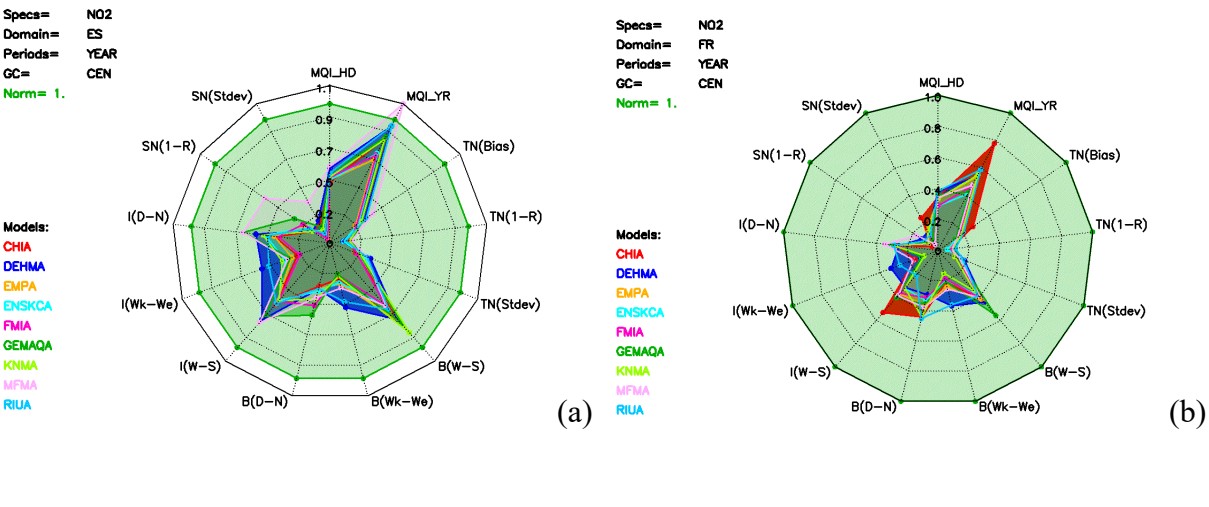

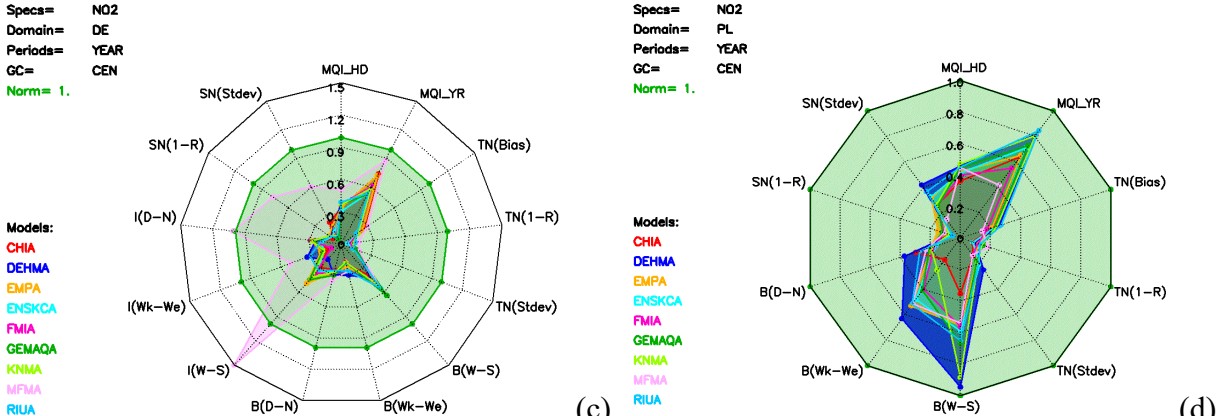

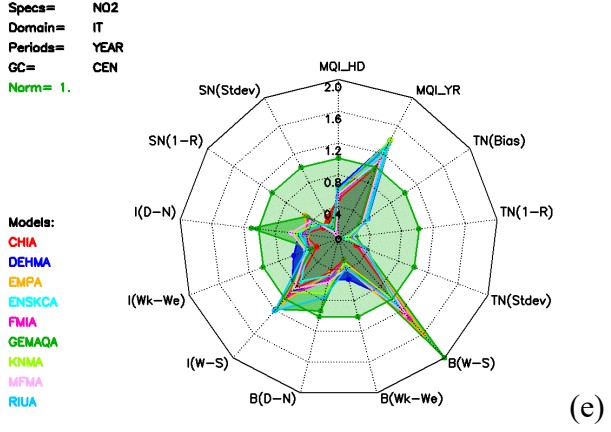

**Fig. 5 Radar plots of the calculated air quality model indicators for NO₂ for different countries excluding the Traffic stations: (a) Spain, (b) France, (c) Germany, (d) Poland and (e) Italy.**

When Traffic stations are excluded from the analysis (Fig. 5), we see that the yearly MQI are much lower for the five countries and even fulfil the MQO for France, Germany and Poland.

This confirms that the models have difficulties in calculating the NO₂ concentrations for Traffic stations. The reason for this is that the model resolution is not fine enough to capture the traffic emissions. The short lifetime of NO₂ (about one hour) requires high model resolution to capture well the non-linear production and loss of NO₂ concentrations.

As indicated, this result was expected and demonstrates that the level of stringency of the QA/QC indicators is relevant. Apart this expected result for traffic stations, these indicators also flag some aspects that need to be improved for $NO_2$, such as the spatial concentration gradient.

All the results of the statistical analysis for $NO_2$ (and other air pollutants) are provided in Table S2 of the Supplement material.

**3.2    Model performance analysis for $PM_{10}$**

The MQI_YRs for $PM_{10}$ concentrations are higher than the MQI_HDs (Fig. 6), which can be explained by the smaller measurement uncertainties for yearly $PM_{10}$ observations as mentioned before. For Germany the Ensemble MQI_YR is close to unity, i.e. 1.00 (± 0.14).

Looking at the different statistical indicators in the radar plots, we see that all the models show similar shapes in the radar
plots, indicating that the models show the same strengths and weaknesses.

The normalized temporal correlation coefficient is expressed in terms of 1-R; the threshold for this indicator remains 1 as for all indicators, meaning that values below 1 fulfil the objective. Values closer to zero indicate even better performances.

This implies that other indicators are required to perform a more stringent evaluation of the ACTM.
The radar plots show that the models have in general difficulties in calculating the spatial profiles (Year UT-UB, UB-RB) and temporal profiles (Winter - Summer gradient for Traffic, Background and Industry) for Spain, France, Poland and Italy. While for Germany all indicators are below unity by the different models, apart from UT-UB and UB-RB by DEHMa and EMPa, and MQI_YRs by DEHMa, GEMAQa and MFMa.

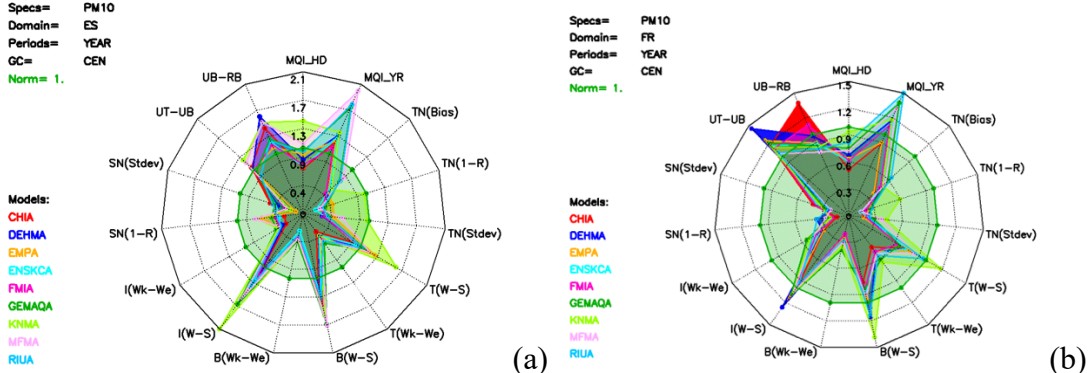

(a)          (b)

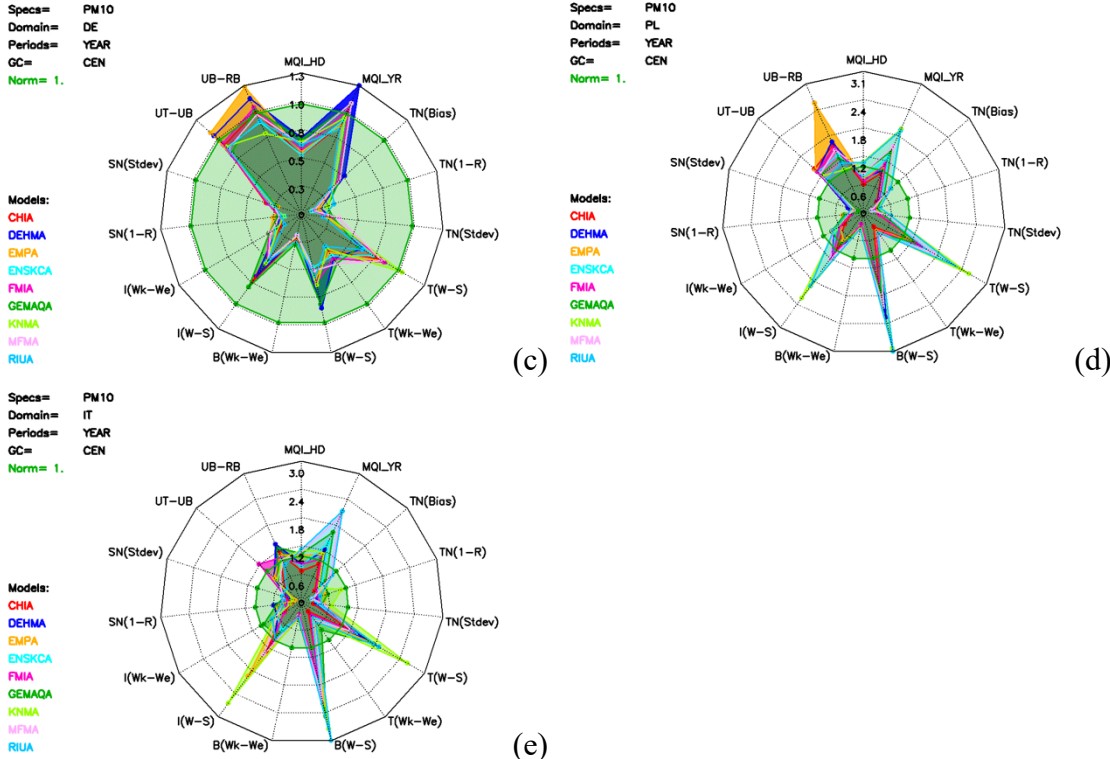

Figure 6. Radar plots of the calculated air quality model indicators for PM$_{10}$ for different countries: (a) Spain, (b) France, (c) Germany, (d) Poland and (e) Italy. Indicators are: MQI Hourly (MQI_HD), MQI Year (MQI_YR), Bias, 1-R (Time), Standard deviation (Time), gradients for Winter-Summer, Week-Weekend, Day-Night for Traffic, Industry, Background (T, I, B), 1-R Spatial, Standard Deviation spatial, Yearly Urban-Traffic vs Urban-Background (Year UT-UB), Yearly Urban-Background vs Rural-Background (Year UB-RB).

The poor skill for Spain and Poland is illustrated in Fig. 7, which shows the large differences between the models in calculating the average PM$_{10}$ concentrations for the different station types. Only DEHMA shows a small positive bias (~1 µg/m$^3$) for all the station types for Spain, while most of the models underestimate on average the observed PM$_{10}$ concentrations.

For Poland, all the models underestimate the observed PM$_{10}$ concentrations for the different station types (Fig.7). The highest PM$_{10}$ concentrations are observed for Traffic stations for Poland. It is for these stations that the models' capability in calculating elevated PM$_{10}$ concentrations for Traffic stations is poor, which is shown in the largest bias found for these stations. Excluding the traffic stations from the comparison results in an MQI of 0.99, while with traffic stations MQI is 1.32.

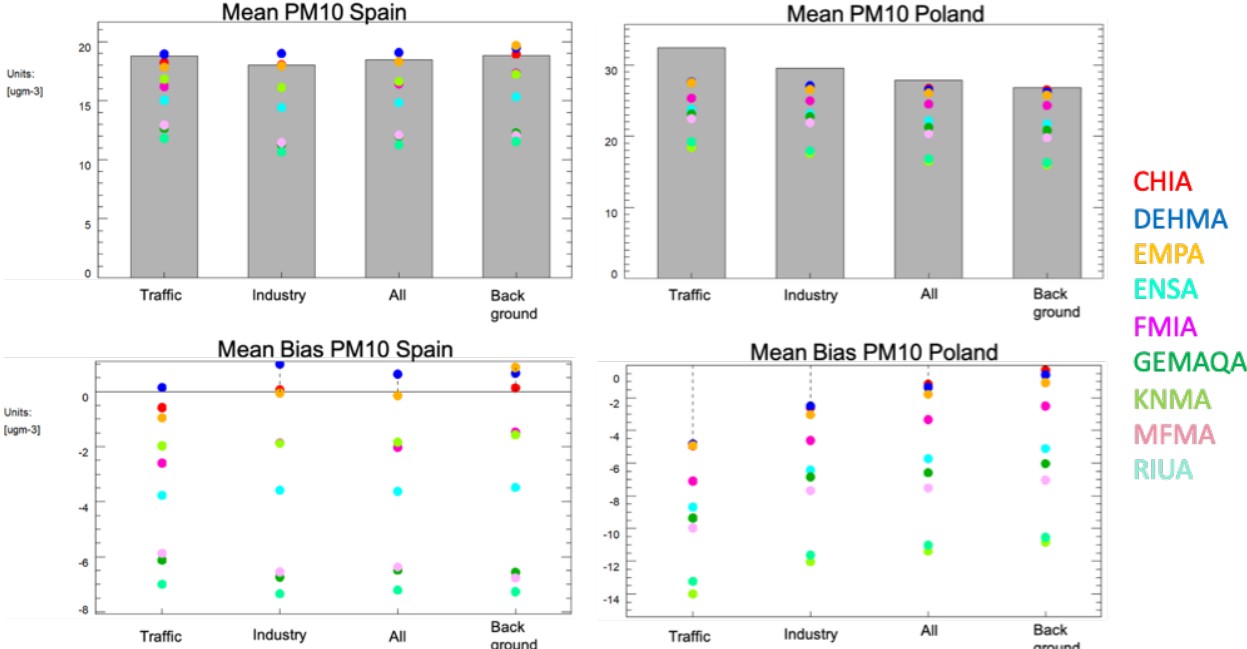

**Figure 7. Mean calculated PM₁₀ concentrations by the 9 models (indicated with coloured bullets) for the different measurement stations (grey bars for Traffic, Industry, All and Background) for Spain and Poland. Together with the bias.**

The radar plots show that the Winter – Summer gradient are larger than 1.0 for the different countries. For that reason, we analyse in more details the PM₁₀ concentrations for Poland during different seasons that will help to understand the reason for the higher bias for traffic. The mean bias during the summer period (Fig. 8, left panel) is the highest for Traffic stations (up to ~-10 µg/m³) with a small positive bias for a few models when All and Background stations are considered. For the winter period (right panel), the mean bias is a factor ~2 higher than for the summer, with RIUA and KNMA showing the highest bias (up to ~-20 µg/m³) for the four different station types. This indicates that the models underestimate the PM₁₀ concentrations for the whole country, especially during winter time, even though the model concentrations are assimilated.

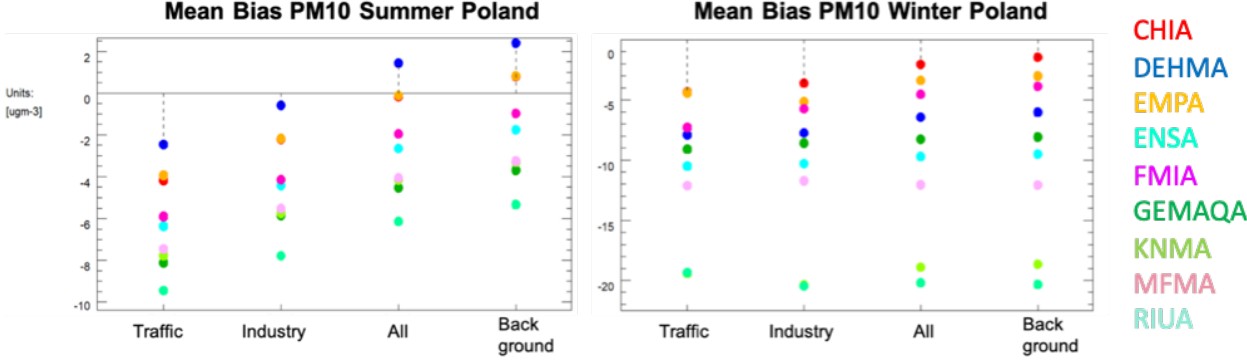

**Figure 8. Mean Bias PM₁₀ for Summer (JJA) and Winter (DJF) for Poland by all the models for the different station types (Traffic, Industry, All and Background).**

When traffic stations are excluded in the analysis, it appears that only for Germany, Poland, and Italy the Ensemble's MQI_YR is lower (e.g. for Poland ~1.4 versus ~1.0 without traffic stations). As mentioned earlier the Winter – Summer profiles for Industry, Background (and to some extend traffic) stations hampers the overall model's performance in

calculation the PM$_{10}$ concentrations (indices are well above the reference criteria of 1.0). For example, the Winter-Summer gradients for Spain (Fig. 9) are scattered around the 1:1 line, while the Week-Weekend profiles are closer to the 1:1 line. The latter corroborates the indicator values below the criteria.

The analysis above tells us that in addition to the MQI, the bias and spatial gradient indicators are relevant and useful to highlight the potential model weaknesses in calculating PM$_{10}$ concentrations. On the other hand, temporal correlation and standard deviation indicators seem to be less useful for evaluating model performance in this context.

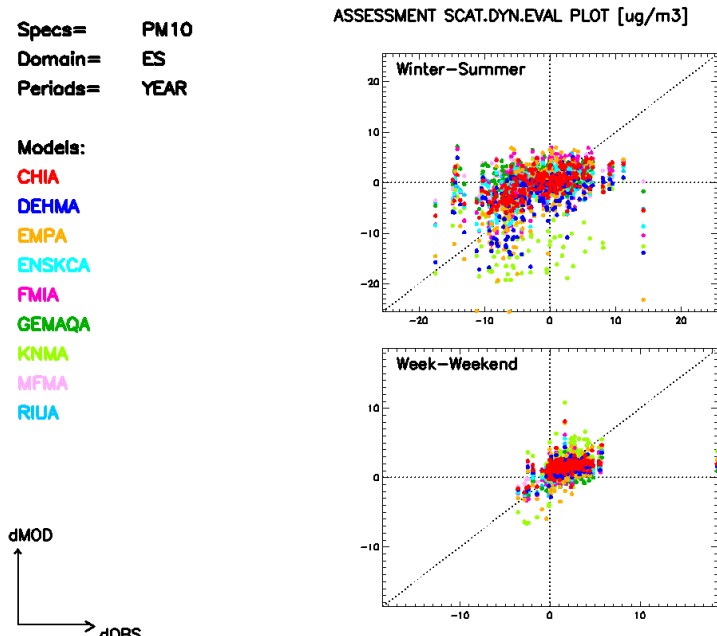

**Figure 9 PM$_{10}$ Scatter plots of modelled versus observed Winter-Summer and Week-Weekend mean differences for Spain for all the models.**

## 3.3     Model performance analysis for PM$_{2.5}$

Yearly MQIs for PM$_{2.5}$ fulfil the MQOs for all models and countries. Also, the MQIs are in general lower than for PM$_{10}$ (Fig. 10). This can be explained by the higher measurement uncertainty assumed for PM$_{2.5}$ than for PM$_{10}$ in the MQI Equations, allowing less stringency on the model results when calculating the MQI for PM$_{2.5}$ (Thunis et al., 2021).

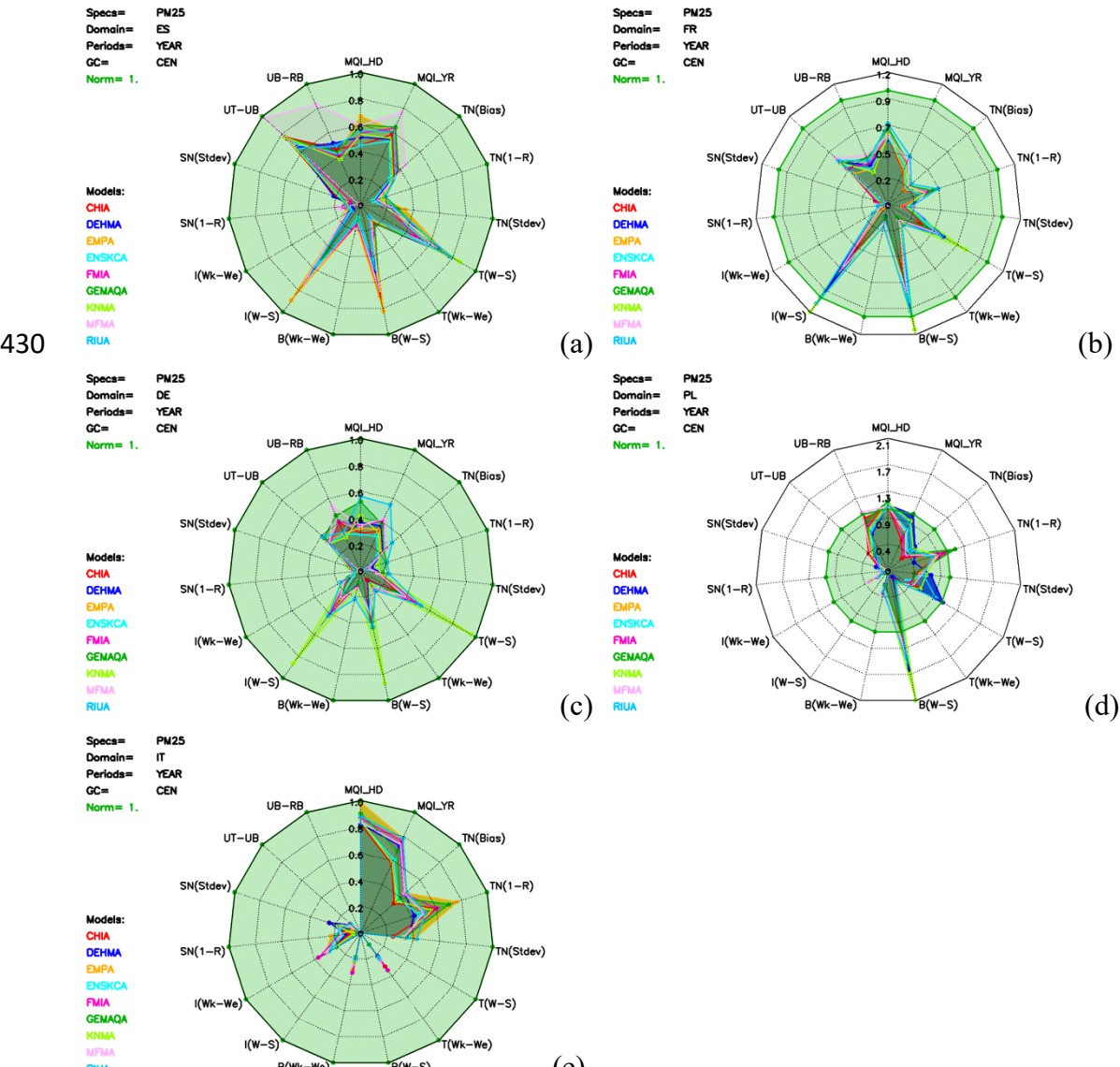

430

**Figure 10. Radar plots of the calculated air quality model indicators for PM$_{2.5}$ for different countries: (a) Spain, (b) France, (c) Germany, (d) Poland and (e) Italy. Indicators are: MQI Hourly (MQI_HD), MQI Year (MQI_YR), Bias, 1-R (Time), Standard deviation (Time), gradients for Winter-Summer, Week-Weekend, Day-Night for Traffic, Industry, Background (T, I, B), 1-R Spatial, Standard Deviation spatial, Yearly Urban-Traffic vs Urban-Background (Year UT-UB), Yearly Urban-Background vs Rural-Background (Year UB-RB).**

440

For Poland where coal combustion in households is still an important contributor to PM (De Meij et al., 2024) larger biases are found for the winter period (up to -13 µg/m$^3$) than for the summer (up to -3 µg/m$^3$), see Fig. 11. Our analysis further showed that for PM$_{2.5}$ Daily and Yearly MQI values for Poland are on average a factor ~2 higher during winter (1.23 and 1.02 respectively) than summer (0.60 and 0.48 respectively). The absence of condensables in the emission

445 inventories (or possibly other seasonal dependent emissions, such as emissions released by forest fires) may lead to much higher biases during the peak season and as a consequence potentially result in higher daily than yearly MQI values.

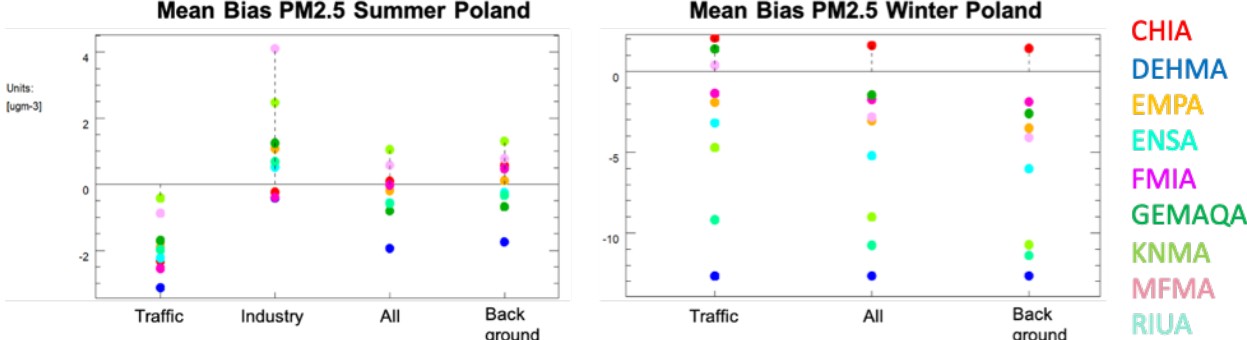

Figure 11. Mean Bias PM$_{2.5}$ for Summer (JJA) and Winter (DJF) for Poland by the models for the different station types (Traffic, Industry, All and Background). Note that for Winter, there's only one Industry station, therefore the bias for this station type is not shown.

As we have seen before, considering only the MQI for the model evaluation doesn't provide enough information of the model's skill in calculating the temporal and spatial variability of the pollutant. The radar plots that include additional temporal and spatial indicators show that for Spain, France and Germany all the models show a similar behaviour, i.e. elevated values for the Winter – Summer indicators for Industry and Background, but still below unity. Just like for Poland, the Winter – Summer profiles for Background, Traffic and Industry stations are higher than 1.0 for DEHMA, KNMA and RIUA. While GEMAQA has difficulties in capturing the temporal correlation.

The analysis raises questions about the stringency of the indicators for PM$_{2.5}$, as passing the criteria does not necessarily indicate flawless performance. The bias and the Winter – Summer indicators reveal potential problems in air quality modelling for PM$_{2.5}$ and for that reason are very useful.

### 3.4    Model performance analysis for O$_3$

For O$_3$, all indicators are lower than unity for France, indicating that the models capture well the 8-hour maximum O$_3$ values (Fig. 12). Except by GEMAQA for Spain, i.e. the Winter-Summer Traffic, Background and Industry indicators are larger than 1.0. This is also true for the Winter-Summer Traffic indicator by RIUa.

Only for Poland, the RIUa model fails to capture the temporal profiles for Winter - Summer for the Traffic and Background stations. Looking in more details at the temporal correlation coefficient (R) for RIUa for all the available stations (35 stations in total), we see that R varies between 0.06 and 0.81 (on average R is 0.63), while for ENSKCa R varies between 0.42 and 0.98 (on average 0.90). This indicates that RIUa has more difficulties to capture the temporal profile for some stations when compared to the other models.

For Italy, MQI_YR is higher than 1.0 by EMPa, FMIa and RIUa, and all the models have difficulties to capture the temporal profile for Winter - Summer Background stations, i.e. the results are scattered around the 1:1 line (not shown). Also, the spatial gradients for UB-RB are higher than 1.0 by GEMAQa and EMPa.

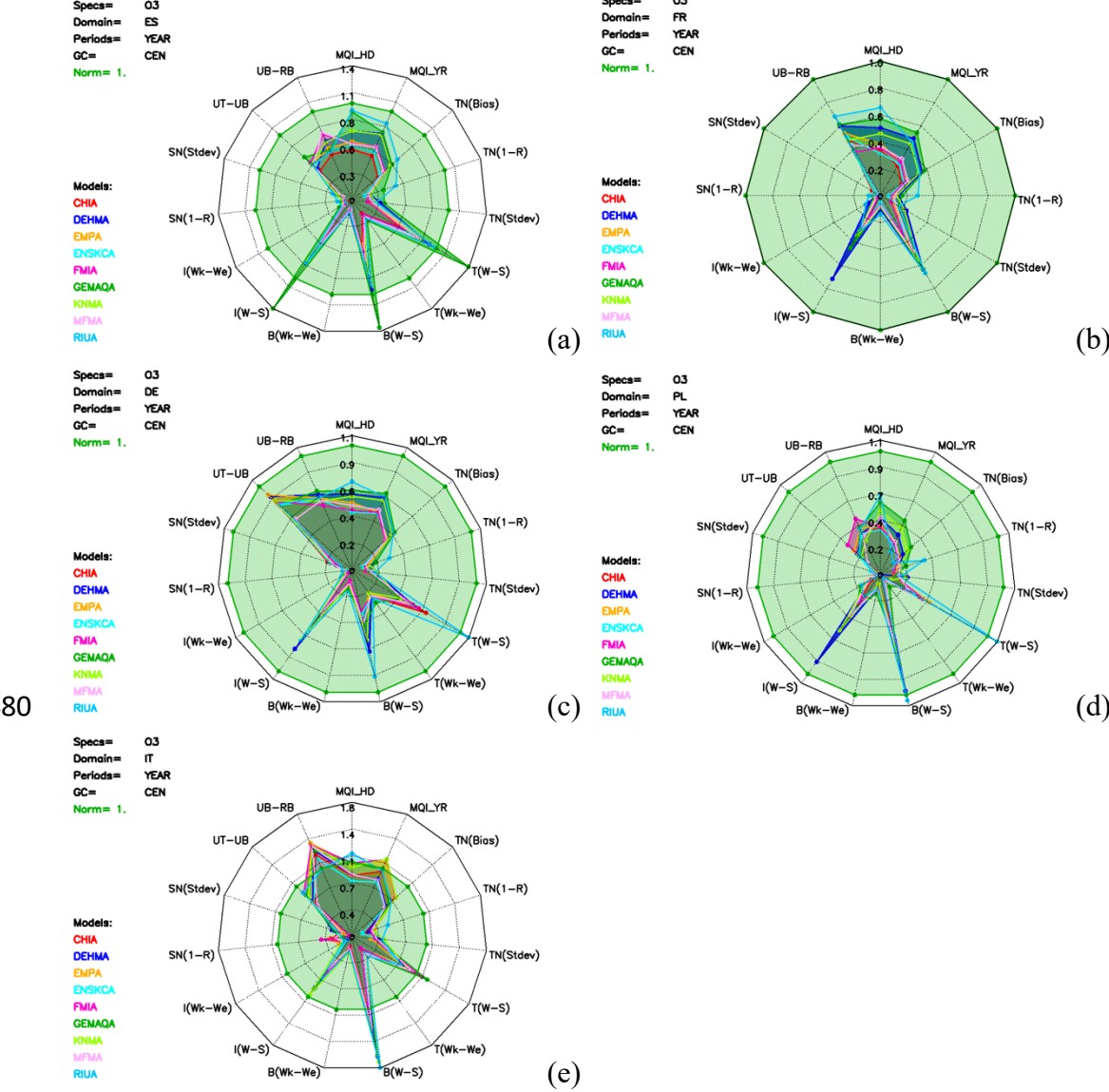

**Figure 12. Radar plots of the calculated air quality model indicators for 8-hour maximum O₃ values for different countries: (a) Spain, (b) France, (c) Germany, (d) Poland and (e) Italy. Indicators are: MQI Hourly (MQI_HD), MQI Year (MQI_YR), Bias, 1-R (Time), Standard deviation (Time), gradients for Winter-Summer, Week-Weekend, Day-Night for Traffic, Industry, Background (T, I, B), 1-R Spatial, Standard Deviation spatial, Yearly Urban-Traffic vs Urban-Background (Year UT-UB), Yearly Urban-Background vs Rural-Background (Year UB-RB).**

Even though the daily and yearly MQI for 8-hour maximum O₃ values are in general below 1.0, the temporal correlation coefficient, together with the Winter-Summer gradients appear to be useful indicators to highlight potential problems for O₃ concentrations modelling.

**4.      Conclusion**

In this work, we examine the relevance and usefulness of assessment indicators within the FAIRMODE framework by evaluating the performance of eight CAMS models and their ensemble in calculating air pollutants. The evaluation is based on comparisons with observations that were not used to assimilate the modelled concentrations.

For nitrogen dioxide ($NO_2$), we found that the yearly Model Quality Indicators (MQI), as well as the Winter-Summer and spatial gradient indicators, clearly show the challenges the models face in accurately calculating $NO_2$ concentrations at traffic stations. This highlights the value of these indicators in assessing model performance. As expected, the exclusion of traffic stations from the analysis improves the models' performance, confirming that the indicators are effectively capturing the models' difficulties. For background stations, all indicator values fall below the threshold of 1.0, except for the GEMAQ model in Italy, suggesting better model performance in less complex environments.

When analysing fine particulate matter ($PM_{2.5}$), we observed that the yearly and daily MQI for all models meet the established criteria. This, however, raises questions about the stringency of the indicators, as passing the criteria does not necessarily indicate flawless performance. Our analysis demonstrated that other indicators, such as bias and Winter-Summer gradients, are crucial for identifying the underlying issues in air quality modelling for $PM_{2.5}$, making these indicators highly valuable.

For $PM_{10}$, the yearly MQI, Winter-Summer indicators, and spatial gradients were not always met by the models. This suggests that, in addition to MQI, bias and both temporal and spatial gradient indicators are particularly important for identifying weaknesses in the models' ability to calculate $PM_{10}$ concentrations. On the other hand, temporal correlation and standard deviation indicators seem to be less useful for evaluating model performance in this context.

Regarding ozone ($O_3$), although the daily and yearly MQI for the 8-hour maximum $O_3$ values generally fall below the threshold of 1.0, additional indicators such as the temporal correlation coefficient and Winter-Summer gradients prove useful for identifying potential model issues in calculating $O_3$ concentrations.

Overall, the various indicators effectively served their purpose of revealing the specific limitations in the model applications, and assisting the modelling community in understanding where improvements are needed. However, there is ongoing debate about the appropriate level of stringency for certain indicators and pollutants, suggesting that there is room for refinement in the evaluation process.

**Code availability**

The IDL (8.8.3) source code of the DeltaToolLight (version 1.4) of the screening method of the statistical analysis can be found here: https://zenodo.org/records/14870503

**Data availability**

The CAMS data is available from the Copernicus CAMS website, via https://atmosphere.copernicus.eu/data .
Also, the observation data and modelling data (at station location) can be found here:
https://zenodo.org/records/14870503

**Author contribution**

ADM performed the data analysis and wrote the draft of the manuscript. CC provided the research tool for the evaluation. CC and PT designed the study and helped with the data analysis. EP collected the data. All co-authors helped in editing
suggestions to the manuscript.

**Competing interests**

The authors declare that they have no conflict of interest.

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
