# Peer review of "A new set of indicators for model evaluation complementing to FAIRMODE's MQO"

_EGUsphere, 2024_

## Author Comment (AC1)

Ispra (Italy), 18 February 2025

Dear Chief Editor,

We have made available the source code and input data that was used to write this manuscript.
You can download the source code and data here:
https://zenodo.org/records/14870503

We hope that this is sufficient to continue the reviewing process of our manuscript " A new set of indicators for model evaluation complementing to FAIRMODE's MQO".

On behalf of all co-authors
Your sincerely,
Alexander de Meij

---

## Author Comment (AC2)

We thank the reviewer for the constructive comments which have been very helpful in improving the manuscript. Please find below a point-to-point reply to the comments.

*Reviewer 1*

*General comments*

*This paper presents a well-structured study and methodology about to evaluate air quality modelling performance, with insightful analysis. The authors have addressed a sistematic analysis of the FAIRMODE work related to this subject and have contributed to produce a reference scientific paper for future air quality modelling applications. However, while the research is robust, there are some minor and major points that should be addressed and corrected.*

*Scientific questions/specific comments*

*- Abstract: The list of indicators tested should be identified previously, in particular the new "increment" indicators that were included in this study. It is not clear what type of indicators were used.*
We added to the text the following to explain better the type of indicators: "Model Quality (bias) and Model Performance (temporal and spatial) Indicators.

*- Lines 57-59: a reference is needed here to support this idea*
We added the following references to this part of the text:
Marécal, V., Peuch, V.-H., Andersson, C., Andersson, S., Arteta, J., Beekmann, M., Benedictow, A., Bergström, R., Bessagnet, B., Cansado, A., Chéroux, F., Colette, A., Coman, A., Curier, R. L., Denier van der Gon, H. A. C., Drouin, A., Elbern, H., Emili, E., Engelen, R. J., Eskes, H. J., Foret, G., Friese, E., Gauss, M., Giannaros, C., Guth, J., Joly, M., Jaumouillé, E., Josse, B., Kadygrov, N., Kaiser, J. W., Krajsek, K., Kuenen, J., Kumar, U., Liora, N., Lopez, E., Malherbe, L., Martinez, I., Melas, D., Meleux, F., Menut, L., Moinat, P., Morales, T., Parmentier, J., Piacentini, A., Plu, M., Poupkou, A., Queguiner, S., Robertson, L., Rouïl, L., Schaap, M., Segers, A., Sofiev, M., Tarasson, L., Thomas, M., Timmermans, R., Valdebenito, Á., van Velthoven, P., van Versendaal, R., Vira, J., and Ung, A.: A regional air quality forecasting system over Europe: the MACC-II daily ensemble production, Geosci. Model Dev., 8, 2777–2813, https://doi.org/10.5194/gmd-8-2777-2015, 2015.

And
Copernicus Atmospheric Monitoring Service, Regional Production, Updated documentation covering all Regional operational systems and the ENSEMBLE, Following U3 upgrade, November 2020,
https://confluence.ecmwf.int/display/CKB/CAMS+Regional%3A+European+air+quality+reanalyses +data+documentation

*- Line 63: which indicators thresholds?*
A threshold is associated to each indicator and corresponds to a level of quality that we assume sufficient for the use of modelling to support policy. Since all indicators are normalized by a quantity

proportional to the measurement uncertainty, this threshold is one for all indicators. We removed this mention to threshold in the text as this was not needed there.

*- Line 73: Why focusing on "the following statistical parameters" – this should be explained*
The indicators and modelling criteria described in this study, were defined in the context of FAIRMODE to support the application of modelling in the context of the Air Quality Directive.
Initially, FAIRMODE developed a single model performance indicator: the MQI. While this indicator provides relevant pass/fail test, passing the test does not ensure that modelling results are fit for purpose. This is why additional indicators have progressively been added, in particular to assess how models capture temporal and spatial aspects.
We added this to the text.

*- Line 82: What is a "complete time series"?*
In our work, a complete time series entails 75% data availability over the selected time period. Note that this number is less than the one requested in the AAQD (90%) to increase the available number of measurement stations for validation. We however impose that available data are representative of the full year. We added this to the text.

*- Line 96: The paper was submitted after the new AQDiretive enter into force, so it should be mentioned*
Indeed, we now refer to the new AADD in the text.

*- Line 102: "temporal or spatial correlation": it shouldn't be "and" instead of "or"?*
The reviewer is right. Indeed, the spatial and temporal indicators are based on temporal or spatial correlation. One indicator is not based on both at the same time.
Thank you.

*- Page 6: stations are only considered in terms of influence, and what about environment type (urban, suburban, rural)?*
Thank you for pointing out this important issue on station types.
In our work, we follow the definitions provided in the Air Quality Directive (2008/50/EC) and the new Ambient Air Quality Directive (Directive 2024/2881/EU) of the European Commission. These definitions are given for different types of air quality monitoring stations based on their location and the pollution sources they are exposed to.

We use mostly the urban types to identify the most important behaviours in air pollutant concentrations. The reason for this is that we believe that there are more important differences between station types than station environments.

*- Page 276: (1-R) lower than 1 do not mean that "models are good for these indicators", only if it close to zero... and also, "good for these indicators" is not the most appropriate scientific expression to evaluate model skills*

We rephrased the sentence which now reads as follows:

The normalized temporal correlation coefficient is expressed in terms of 1-R; the threshold for this indicator remains 1 as for all indicators, meaning that values below 1 fulfill the objective. Values closer to zero indicate even better performances.

*- More general comment: the analysis of results (section 3) is mainly focused on the behaviour of each model in each country/area/pollutant, which I don't think is the main goal of the paper/study. Only in the conclusions section is discussed the main question: how the different indicators are useful and should be used for each different pollutant. The ideas that are written in the conclusions should be already addressed and presented before, during the analysis of the results - that should be always focused on the indicators.*

We agree with the reviewer. However, while the country analysis does not address the main question of the usefulness of the indicators, we nevertheless need this analysis to assess how these indicators behave across Europe. We went through the manuscript and tried to stress these points where relevant and useful.

*Technical corrections:*

*- Using different terminology ("Air Chemistry Transport Models (ACTMs)" and "Air quality models" can be confusing.*

We have made the corrections in the text where appropriate. Thank you.

*- The number of athoms in the chemical formula of the pollutants (NO2, O3) should be subscript*

Corrected.

*- Line 50: Replace model by models on "More details on the model"*

Done.

*- Line 58: 0.1x0.1 is approximately 11km (and not 10km)*

We disagree. The distance in kilometers between two longitude points differ when moving away from the North Pole, i.e. the distance is getting larger. Depending on the position on the Earth, the distance in kilometers varies. For example, 0.1 degrees longitude around Tromso (Norway) is around 3.8 km, while 0.1 degrees longitude around Athens is ~8.9 km. The distance of 0.1 degrees longitude at the equator is around 11.1 km.

We removed "approx. 10 km" to avoid confusion.

*- Line 62: replace "calculated" by "simulated"*

Done.

*- Line 90: add the symbol of the mean measured concentration "(U(O)):"*
Done.

*- Page 3, Line 65: avoid the 2 words together "simulated calculated"*
Corrected. Thank you.

*- Line 127: "relevant1"?*
The number 1 refers to the first footnote in the manuscript. To make the meaning of 1 clearer, we have placed it as a superscript.

*- Page 5: tables are not numbered*
Done, thank you.

*- figures in pdf do not show good quality*
We've tried to improve the quality of the figures.

*- Line 376: senteces whould never start with "While"*
We disagree with the reviewer. The Cambridge Grammar of the English Language (Huddleston & Pullum, 2002) confirms that "while" can be used at the beginning of a sentence.

---

## Author Comment (AC3)

We thank the reviewers for the constructive comments which have been very helpful in improving the manuscript. Please find below a point-to-point reply to the comments.

*Reviewer 2*

*This paper assesses the relevance and usefulness of the model performance indicators developed within the FAIRMODE framework by evaluating 8 CAMS models and their ensemble results for predicting four major air pollutants (NO2, O3, PM2.5 and PM10) across Europe. The study compares the model predicted air pollutant concentrations with observations, and highlights the limitations of the current MQOs and the need to reconsider the strictness of some indicators for certain pollutants. The major limitation of the current MQOs is that they provide a single pass/fail summary for a modelling application, which allows a modelling test to pass for the wrong reason under certain circumstances.*

*Additionally, it does not provide any information on the capability of the model to reproduce spatial variability or on the timing of the pollution peaks. With these in mind, the authors propose a new set of indicators to assess the capacity of models to capture the temporal and spatial variability, complementing the current FAIRMODE MQOs. While the manuscript makes a valuable contribution to model performance evaluation by proposing more comprehensive indicators, I have several concerns for the authors to address before the manuscript can be considered for publication.*

*Major concerns:*
*1. The methodology section requires more detailed information. Key aspects such as the emissions inventory, meteorological simulations, modelling time period (winter? Summer? 2021 whole year?), modelling domain, and parameterizations of the models should be described. These details should at least be included in the supplementary information and briefly mentioned in the main text to help readers understand the origins of uncertainties. It would be helpful to include a brief discussion of the assumptions made during model construction and any limitations of the current approach.*

We've added the following text to the manuscript in section 2 and put the table below in the Supplement material.

"The CAMS regional air quality models generate reanalysis, detailing the concentrations of major atmospheric pollutants in the lowest layers of the atmosphere across the European domain (ranging from 25.0°W to 45.0°E and 30.0°N to 72.0°N). The horizontal resolution is approximately 0.1°, varying from around 3 km at 72.0°N to 10 km at 30.0°N. Uncertainties in the representation of dynamical and chemical processes, emission inventories and meteorological input data typically limit the accuracy of calculated gas and aerosol concentrations (De Meij et al., 2012 and references therein). For that reason, an overview of the type of assimilation methodology, which species are assimilated, together with gas and aerosol schemes are given in Table S1 of the Supplement material. More details of the different models are described in (https://confluence.ecmwf.int/display/CKB/CAMS+Regional%3A+European+air+quality+reanalyses+data+documentation)."

**Supplement material**
**Table S1 Overview model characteristics**

| Model | Meteorological driver | Emissions | Boundary Conditions | Gas phase chemistry / Inorganic aerosols | Assimilated surface pollutants | Assimilation |
|---|---|---|---|---|---|---|
| Chimere | IFS, 3 | CAMS-REG-AP | CAMS- | MELCHIOR 2 / | NO2, O3, | Kriging- |

| | | | | | | | |
|---|---|---|---|---|---|---|---|
| | hourly | | | Global IFS | ISORROPIA 2.1 | PM2.5, PM10 | based |
| DEHM | IFS, hourly | 3 | CAMS-REG-AP | CAMS-Global IFS | Modified Strand and Hov (1994) / Frohn (2004) | NO2, CO, SO2 O3, PM2.5, PM10 | Intermittent 3D-Var |
| EMEP | IFS, hourly | 3 | CAMS-REG-AP | CAMS-Global IFS | EmChem19a / MARS (Binkowski and Shankar, 1995) | NO2, CO, SO2, O3, PM2.5, PM10 | Intermittent 3D-Var |
| EURAD | IFS | | CAMS-REG-AP | CAMS-Global IFS | RACM-MM/ Thermodynamic equilibrium (Friese and Ebel, 2010) | NO2, CO, SO2 O3, PM2.5, PM10 | Intermittent 3D-Var |
| GEMAQ | IFS, hourly | 3 | CAMS-REG-AP | CAMS-Global IFS | Modified ADOM IIB mechanism / Gong et al., (2003) | NO2, O3, PM2.5, PM10 | Optimal Interpolation |
| Lotos Euros | IFS, hourly | 3 | CAMS-REG-AP | CAMS-Global IFS | Modified CBM-IV / ISORROPIA-2 | NO2, O3, PM2.5, PM10 | Zhang (2001) |
| MOCAGE | IFS, hourly | 1 | CAMS-REG-AP | CAMS-Global IFS + MOCAGE | RACM / ISORROPIA-2 | NO2, O3, PM2.5, PM10 | 3D-Var |
| SILAM | IFS, hourly | 1 | CAMS-REG-AP | CAMS-Global IFS | CBM-IV / Sofiev (2000) | NO2, O3, CO, SO2, PM2.5, PM10 | Intermittent 3D-Var / |

Binkowski, F. and Shankar, U.: The Regional Particulate Matter Model .1. Model description and preliminary results, J. Geophys. Res., 100, 26191–26209, 1995.

De Meij, A., Pozzer, A., Pringle, K. J., Tost, H., and Lelieveld, J., EMAC model evaluation and analysis of atmospheric aerosol properties and distribution, Atmos. Res., 114–115, 38–69, 2012.

Friese E, Ebel A. Temperature dependent thermodynamic model of the system $H(+)-NH_4(+)-Na(+)-SO_4^{2-}-NO_3^{-}-Cl^{-}-H_2O$. J Phys Chem A. 2010 Nov 4;114(43):11595-631. doi: 10.1021/jp101041j. PMID: 21504090.

Frohn, L. M.: A study of long-term high-resolution air pollution modelling, Ministry of the Environment, National Environmental Research Institute, Roskilde, Denmark, 444 pp., 2004.

Gong, S. L., Barrie, L. A., Blanchet, J.-P., von Salzen, K., Lohmann, U., Lesins, G., et al. (2003). Canadian aerosol module: A size-segregated simulation of atmospheric aerosol processes for climate and air quality models 1. Module development. Journal of Geophysical Research, 108(D1), 4007. https://doi.org/10.1029/2001JD002002.

Sofiev, M.: A model for the evaluation of long-term airborne pollution transport at regional and continental scales, Atmos. Environ., 34, 2481–2493, 2000.

Strand, A., and Hov, Ø.: A two-dimensional global study of tropo- spheric ozone production, J. Geophys. Res., 99, 22877–22895, 1994.

*2. The manuscript allocates receptors to categories including background, urban, traffic and industry. Does the current classification fully capture the diversity of the environments? A clear definition of what each category (e.g., "traffic," "industry") represents is needed, along with*

*justification for why these specific categories were chosen. In my mind, urban areas often exhibit both traffic-related pollution and residential zones, what's the difference between "urban" and "traffic"? Does "traffic" mean receptors adjacent to road, while "urban" refers to receptors away from road but in urban residential area?*

Thank you for pointing out this important issue on station types.

The Air Quality Directive (2008/50/EC) and the new Ambient Air Quality Directive (Directive 2024/2881/EU) of the European Commission provides definitions for different types of air quality monitoring stations based on their location and the pollution sources they are exposed to. These station types ensure a comprehensive assessment of air quality across different environments, helping policymakers and researchers analyze pollution trends and enforce regulatory limits. We use mostly the urban types to identify the most important behaviours in air pollutant concentrations. The reason for this is that we believe that there are more import differences between station types than station environments.

The key definitions are:

A **traffic station** is located in areas where pollution levels are significantly influenced by emissions from road traffic. These stations are typically placed:

- Near major roads, highways, or intersections.
- Where vehicle emissions (such as $NO_2$, PM10, PM2.5) dominate the air quality levels.
- In locations ensuring that they reflect the exposure of the population to pollution from road transport.

An **urban station** represents the overall air quality in an urban area without being directly affected by a specific pollution source like traffic or industrial emissions. These stations are:

- Located in residential, commercial, or mixed areas.
- Reflecting the exposure of the general urban population.
- Measuring background pollution levels influenced by a mix of sources.

**Industrial stations** are located near significant industrial sources, such as factories or power plants. The stations:

- Monitor emissions from industrial activities and their impact on surrounding areas.
- Typical pollutants: $SO_2$, $NO_2$, heavy metals, VOCs.

A **rural station** is placed in areas away from direct local pollution sources, representing regional air quality. These stations:

- Measure background pollution levels from natural and transboundary sources.
- Are located in the countryside or suburban areas far from significant emissions (e.g., cities, industrial areas, or major roads).
- Help assess long-range transport of pollutants.

The Air Quality Directives provides detailed criteria for air quality monitoring station. Below are the definitions and references to the relevant sections given:

1. Traffic Stations

These stations measure pollution primarily from road traffic and are located where the highest concentrations of pollutants due to traffic emissions are expected.

They should be at least 25 meters from major intersections but no more than 10 meters from the road. They must be positioned to represent the population's exposure to pollution from traffic.

2. Urban Background Stations
These stations measure general air quality in urban areas without direct influence from traffic or industry. They must be more than 50 meters away from major roads and more than 4 km away from industrial sources. Their purpose is to assess the average exposure of the urban population to air pollution.

3. Rural and Suburban Background Stations
These stations are located in areas with minimal direct pollution sources, representing the regional or background air quality. Rural stations are placed at least 20 km from urban areas and 5 km from industrial sources. Suburban stations can be closer to cities but should not be influenced by local sources.

We summarized the above information regarding the station types and added this to the manuscript in Section 2.

*3. The current FAIRMODE MQOs considers four air pollutants including NO2, O3, PM2.5 and PM10, why don't the authors include more air pollutants such as SO2, CO, and PM2.5 chemical species? Additionally, the paper considers 8-hour maximum O3 values, how about 1-h max O3 peaks?*
In this study we selected $NO_2$, $O_3$, $PM_{2.5}$ and $PM_{10}$ to investigate the usefulness of the indicators. It is important to note that building a MQI for one pollutant and time aggregation requires information on the associated measurement uncertainty. This is not straightforward to obtain. This is why we focused on the four main pollutants and for each only considered one short and one long time aggregation. Work is currently ongoing to extend these MQI to additional pollutants and time averages.

*4. Given the complexity of air quality modelling, including an uncertainty analysis or a discussion of the confidence in the model's predictions would be valuable. This would provide more insight into the reliability of the proposed indicators and how they could be applied in practice.*
The Reviewer has a valid point.
We are not sure to understand your point but here is an explanation of what we try to achieve with our approach. Estimating the modelling uncertainty is almost impossible, as it would require a large number of model simulations where each parameter is modified independently. Given this difficulty, we assume in our approach that the modelling uncertainty is proportional to the measurement uncertainty. The more uncertain the measurement, the more flexibility we allow to the model results. This coefficient of proportionality is obviously challenging to fix. It should lead to a threshold that is sufficiently stringent to ensure sufficient quality but not too stringent that no model fulfills it. The sensitivity analysis consists in selecting a large number of model simulations and test them against different threshold levels to identify the relevant level of stringency. Our work constitutes one test in this context but more tests will be performed in future.

*5. After introducing the new set of indicators, it would be helpful to provide a full table summarizing the complete set of MQO indicators. Comparisons with other well-established model performance indicators from different regions (e.g., the US, China, and India) are also necessary. This would provide a more comprehensive evaluation and context for the proposed indicators.*

As suggested by the reviewer we included the use of model performance indicators applied in other regions in the world and placed this is a new section Discussion. We added the following to the section 
[revised manuscript text omitted]

""

*Specific Comments:*
*6. Please ensure that the use of subscripts and superscripts for air pollutants and other variables is consistent throughout the manuscript. For example, NO2 should be NO2; μg/m3 should be μg/m3.*
Corrected.

*7. On page 4, line 93, the MQO is first mentioned, but its definition is provided later in line 99. The abbreviations should be defined at the first time it appears.*
Corrected.

*8. I recommend adding a more detailed explanation of the variables used in each formula. Many variables in the manuscript are not clearly defined, which could lead to confusion for readers. A thorough description of each term will enhance the clarity of the model formulations.*
A detailed description of each variable addressed in this study is provided in Janssen et al, (2022). This is also mentioned in the manuscript. We believe the descriptions of the variables are sufficient, keeping in mind that the goal of this work is to evaluate the usefulness of the variables. Detailing each variable would make the manuscript become unnecessary lengthy.

*9. All tables in this manuscript are missing table numbers, titles or captions. Please provide clear titles for all tables to give context to the data being presented.*
Corrected. Thank you.

*10. Section titles with a single variable name (e.g., "NO₂") do not provide enough information about the content of the section. I suggest adding brief summaries to section titles to help readers understand the focus of each section.*
Done.

*11. In some radar plots, the brackets around serial numbers are partially obscured, and some incomplete solid lines extend outside the borders of other figures. These issues detract from the overall appearance of the figures and should be corrected to improve the presentation.*
Corrected.

*12. On page 8, line 207, the phrase "for Traffic, Industry, All and Background stations for Germany" is unclear. What is meant by "All stations"? Is this the sum of traffic, industry, and background stations? If so, why does Figure 2 show lower NO₂ concentrations at all stations compared to traffic stations? This requires further clarification.*
Average of all station types considered. And the reason why the $NO_2$ concentrations are lower for "All stations", is that also the background concentrations are considered. Note that the number of stations for each station type (urban, traffic, industry) also differs, which affects the $NO_2$ concentrations when all stations are considered.

*13. The font size within Figure 4 varies, which impacts the readability and visual quality. I recommend enlarging the font size to improve consistency and clarity.*
We have corrected the font size to enhance the readability of the figure where applicable.

*14. Line260, "The reason for this is that the model resolution is not fine enough to capture the traffic emissions and as a result the short lifetime of NO2 (about one hour) and consequently the non-linear production and loss of NO2 concentrations." suggests a direct causal relationship between model resolution and the short life of $NO_2$. This could be misleading; I recommend rephrasing to avoid suggesting that insufficient model resolution directly impacts the short lifetime of $NO_2$. The two phenomena are not causally linked in this manner.*

As suggested by the reviewer we rephrased the sentence. It now reads as follows:

"The reason for this is that the model resolution is not fine enough to capture the traffic emissions. The short lifetime of $NO_2$ (about one hour) requires high model resolution to capture well the non-linear production and loss of $NO_2$ concentrations."

*15. Line 277, the word "that" is duplicated in the sentence.*
Corrected.

*16. The Conclusion section primarily summarizes the findings but does not delve into a deeper discussion or implications of the results. I suggest expanding this section to discuss the broader implications of the proposed indicators, including how they could influence model evaluation in other regions or in future air quality studies.*

Initially, FAIRMODE introduced a single model performance indicator, the MQI. While this indicator provides a relevant pass/fail test, passing the MQI does not necessarily guarantee that the modeling results are fit for purpose. To address this, additional indicators have been progressively introduced, particularly to assess how models capture temporal and spatial aspects. At this stage of evaluating the usefulness and relevance of these indicators, we analyzed five countries and three air pollutants to better determine whether a given indicator is useful and relevant for a specific pollutant. The methodology presented in this study will be applied to a broader range of air pollutants and countries in the future. Also, our methodology could be applied in other regions in the world where some model performance indicators are already used, like the EPA in the USA and in China to enhance the robustness of the modelling air quality results.

*17. There are several typographical errors throughout the manuscript (e.g., "u" should be "μ"). A careful proofreading is required to correct these and improve the manuscript's overall quality.*
Corrected.

---

## Author Comment (AC4)

We thank the reviewer for the constructive comments ,which have been very helpful in improving the manuscript. Please find below a point-to-point reply to the comments.

***Reviewer 3***

*The authors expose the strategy developed in the framework of FAIRMODE in order to qualify the performance of model outputs, starting with the specific case of the CAMS modelling framework (for models), and the Airbase network for measurements. Specifically, they present a new set of indicators to evaluate more thoroughly the performance of air quality models in the framework of the European CAMS ensemble simulation. This new set of criteria permit to evaluate not only the general performance of the model in standard statistical fashion, but develops new metrics to focus on specific features such as the weekly, diurnal and seasonal cycle, and spatial differences between different station types..*

*The matter of this article (improving and complementing the set of criteria and metrics used to benchmark model performance) is of interest and seems timely. However, I have strong concerns. The bibliography of the article is almost non-existent (five studies are cited, including 4 by the same authors as this papers), highlighting the fact that the proposed method is not compared to other efforts in other countries, with other approaches. In my opinion, it is impossible to publish a research article without placing the work in the context of the international state-of-the-art.*

*Also, the performance criteria are based essentially on measurement uncertainty, a possibly interesting approach, very different from what is done elsewhere, but the authors do not discuss their efforts in light of other existing strategies, reducing the scientific interest of the paper. The authors spend most of the time in the manuscript to showcase the application of these new criteria for validation of model outputs on specific European countries (Spain, France, Italy, Germany and Poland), but without really discussing the methodological basis for these criteria, and how their criteria differ (or improve upon) other methodologies. In this respect, it seems to me that the present paper is designed more like an internal technical report rather than a scientific paper presenting criteria intended to be used by others, and compared to the production of others.*

*Therefore, I recommend rejection of this article. Since the matter is of interest, I recommend a new submission of a totally revised and reoriented manuscript focused on discussing the design of the criteria and placing the methodology of the authors in a wider context.*

We accept that there is room for improvement, and inserted in the manuscript a new chapter Discussion, which provides an overview of previous work that apply Model Performance Indicators and Criteria in the USA and China. This has also led to the extension of the bibliography.

[revised manuscript text omitted]